# Engaging Employers in Apprentice Training: Focus Group Insights from Small-to-Medium-Sized Employers in Ontario, Canada

**DOI:** 10.3390/ijerph20032527

**Published:** 2023-01-31

**Authors:** Aaron S. Howe, Joyce Lo, Sharan Jaswal, Ali Bani-Fatemi, Vijay Kumar Chattu, Behdin Nowrouzi-Kia

**Affiliations:** 1Department of Clinical and Counselling Psychology, Columbia University, New York, NY 10027, USA; 2Restore Lab, Department of Occupational Science and Occupational Therapy, University of Toronto, Toronto, ON M5G 1V7, Canada; 3Department of Community Medicine, Faculty of Medicine, Datta Meghe Institute of Medical Sciences, Wardha 442107, India; 4Center for Transdisciplinary Research, Saveetha Dental College, Saveetha Institute of Medical and Technical Sciences, Saveetha University, Chennai 600077, India

**Keywords:** apprenticeship, mental health, workplace culture, gender, tradesperson

## Abstract

Several factors have been identified to influence the registration and retention of apprentices in the construction trades. Employer engagement is a key factor to promote growth in apprenticeships in the construction trades as participation rates continue to be low among small-to-medium-sized employers. In this study, we evaluated the effectiveness of the Ontario Electrical League’s (OEL) employer mentorship program through the perspectives of small-to-medium-sized employers using a qualitative approach. Two focus groups were conducted virtually with 11 employers. Focus group audio transcripts were recorded and transcribed for thematic analysis. Themes were generated using a data-driven approach to examine the relationships between mentorship program outcomes and perspectives on industry-related recruitment and retention barriers. Three themes were identified: (a) long-term apprentice recruitment and retention challenges; (b) equity and mental health in the workplace; and (c) industry challenges and mentorship program outcomes. Generally, this sample of employers appreciated the value of the OEL mentorship program through praise of the continued educational support, employer management expertise, hiring resources, and apprentice onboarding tools despite industry barriers in trade stigma, equity and mental health in the workplace, and recruitment and retention challenges. Industry partners should work with these small-to-medium-sized employers to develop workplace initiatives and engage external partners to provide ongoing apprenticeship mentorship support to address the recruitment and retention barriers identified in this study.

## 1. Introduction

It is estimated that 88,960 new journeypersons and 296,350 apprentices will be required for the Top 50 Red Seal trades to sustain the current and future demands of the Ontario, Canada, labor market [1]. With this demand, there has been growing pressure for employers to increase awareness and recruit and engage potential apprentices to consider vocational pursuits in the trades industry [2]. These pressures have been compounded by the external influences of economic turbulence (e.g., supply and demand for services, wages), the COVID-19 pandemic, and decreased educational interest in trades apprenticeships for adolescents and young adults over the past five years [1,2,3,4]. From 2016 to 2020, there has been a notable decrease in apprentice registration and completion rates for 12 of the 15 largest Red Seal trades, including construction electricians and plumbers [1,4]. Future projections have suggested a modest recovery of these rates in the coming years, and therefore [1,4], industry partner organizations have examined extrinsic and intrinsic factors that may be influential for sustainable future growth.

The current apprenticeship model in Ontario allows for apprentices to engage in “on-the-job” vocational training under the supervision of a tradesperson [3]. These apprentices are also required to engage in classroom instruction to develop their competencies, in preparation of becoming a licensed tradesperson [3,4]. Apprentices must find an employer that is qualified and agreeable to provide vocational training to commence their apprenticeship, which is regulated by the provincial government [5]. In 2018, Bill 47 shifted the ratios from 3:1 to 1:1 (journeyperson to apprentice) which has been beneficial for increasing the capacity for apprenticeship opportunities but has also reduced the quality of the apprentices available [5]. Apprenticeship ratios have been a long-standing issue for the trades industry in Ontario due to the implications of injuries, delays in licensure, and apprentice capacity for small-to-medium-sized employers (SMEs) [3]. SMEs account for the largest proportion of the Ontario workforce in the skilled trades [6]. A SME is defined as “an enterprise with fewer than 499 employees” [6]. Therefore, increasing apprenticeship opportunities in SMEs may be a potential opportunity to increase recruitment and retention in the skilled trades given that they represent a large proportion of the workforce in the skilled trades. However, SMEs may not have the infrastructure and workplace initiatives to support increased recruitment of apprentices due to costs of training, a lack of administrative support, difficulty managing 1:1 ratios, and the absence of apprentice support programs [3].

Previous reports have suggested that Canadian organizations spend less per capita on training than other industrialized countries (e.g., United States, Australia, Germany) [5]. This is further supported by low participation rates (16–30%) of trades employers in apprenticeship programs [3,4] and an ongoing lack of awareness of the benefits of apprenticeship programs for employers [4,7]. This under-emphasis on training for apprentices can create long-term effects on apprentice completion rates and retention. A secondary intrinsic factor that has received ongoing attention is the lack of diversity in the construction trades. Those that are currently underrepresented in the construction trades (e.g., persons with disability, women, Indigenous persons) have been disproportionately affected in the labor market by COVID-19 and continue to be untapped potential for increasing the skilled trades labor supply [7,8]. In Canada, estimates suggest that women make up only 2.5% of the construction trades [9,10,11,12,13,14,15]. Those women who do work in the construction trades often occupy off-site jobs that are administrative or non-technical in nature [14,15]. Similar participation rates have been observed in Indigenous persons, at 4.7% in Canada and 2.9% in Ontario (2nd lowest in Canada) [15]. Increasing equitable mentorship and training opportunities amongst these underrepresented groups may be another avenue of growth for the construction industry.

In this study, we have collaborated with the Ontario Electrical League (OEL) to evaluate their mentorship program [16]. The goal of their mentorship program is to engage SMEs to participate in apprentice training by promoting their employer support services, employer business resources, and networking ability to connect SMEs with high-quality apprentices. The OEL is a long-standing, non-profit organization that provides support services for SMEs and resources for equitable apprentice advocacy in their progression through the construction electrical trades apprenticeship program. Through active support in these key areas, the long-term goal of the OEL is to promote stronger SME engagement and advocacy to increase apprenticeship capacity and sustainable growth of the skilled trades industry [16]. The OEL also plays a pivotal role in government lobbying to advocate for funding to further refine non-financial and financial support(s) with these employers to mitigate the initial investment costs of recruiting an early year (1st or 2nd year) apprentice.

The primary aim of this study was to evaluate the effectiveness of the OEL mentorship program through the perspectives of SME employers using a qualitative approach. This aim would allow for the OEL and other advocating skilled trades organizations to further refine current support(s) available and consider innovative offerings to continue to grow their mentorship program(s) by creating additional opportunities. Evidence-based analysis of our findings will be organized into preliminary recommendations for the OEL and the skilled trades industry in Ontario. The secondary aim was to identify recruitment barriers or challenges in the current apprenticeship model for electrical SMEs and their apprentices. The stable decrease in persons pursuing a career in the skilled trades over the last decade has parlayed into an ongoing shortage of skilled trades workers to meet current and future demands. Therefore, we engaged the perspectives of SME employers and mentors to understand how to further recruit and integrate underrepresented (e.g., females, ethnic minorities/newcomers to Canada, Indigenous persons) and disadvantaged persons (e.g., persons with disabilities) in the construction trades industry.

## 2. Materials and Methods

This qualitative study was performed in Toronto, Ontario, Canada, as a part of a larger exploratory, sequential mixed methods cross-sectional design. Focus groups were conducted with SME electrical employers to evaluate the outcomes and effectiveness of the OEL mentorship program. Eligible employers were screened through a registry maintained by the OEL and were required to have between 0 and 50 employees to qualify as a small-to-medium-sized employer. Screened employers were invited to participate in a quantitative survey, and those who completed the survey were asked for voluntary interest in a focus group. These exploratory findings were used to inform internal and external stakeholders of influential factors that could be used to promote SME engagement and reduce barriers-to-entry for newcomers and underrepresented persons to the skilled trades. They also were used to determine feasibility for large-scale studies to further identify mental health outcomes (e.g., burnout/work-related stress, emotional distress) and the effects of employer workplace culture on diversity, equity, and inclusion in the skilled trades.

This research study was conducted in collaboration with the OEL and as part of the OEL’s “Increasing Employer Engagement in Apprenticeship Training” 12-month mentorship project to meet the current and future needs of the skilled trades labor market [16]. In this project, the OEL provided employer support and apprenticeship advocacy through small-group mentorship discussion sessions, employer outreach and marketing, education/information manuals, training and hiring tools, and coordination of services with industry partners.

### 2.1. Focus Group Questions

The focus group questions were developed collaboratively with the OEL [17]. The researchers generated a series of preliminary questions and worked closely with the OEL to finalize the relevance and importance of each question in evaluation of the mentorship program. Participants were asked about their work experiences and key questions regarding their perceptions of the outcomes and effectiveness of the OEL mentorship program. Questions were asked as follows: “What do you think about the apprenticeship program; How would you describe OEL’s support in apprentice training; What do you like about the OEL mentoring/training program; What are aspects of the OEL mentoring/training program that you would like to see changed; Tell me about some of the challenges facing employers in training an apprentice; What type of support do you need to provide apprenticeship training opportunities and ensure Ontarians have the skilled trades workforce they need; Has your mental health been affected while working in the skilled trades during the COVID-19 pandemic; What are other strategies do you think OEL should consider to improve employer engagement and was the program successfully encouraging you to hire apprentices from underrepresented groups in the labor market or those disproportionately impacted by COVID-19, including women, youth, persons with disabilities, racialized groups, and indigenous peoples?” Mental health in the workplace was defined as a respectful and productive work environment that promotes, preserves, and considers the mental wellbeing of the employees [18].

### 2.2. Participants and Procedure

Two focus groups, 60–70 min in duration, were conducted virtually on Microsoft Teams (version 1.5.0). There were 11 SME electrical employers across Ontario that participated in the two focus groups (n = 11; 5–6 employers in each group). Most of the participants were electrical employers (n = 10, 90.1%) and one was a plumbing employer. All the employers were SME business owners. There were 10 males and 1 female that participated between the ages of 34 and 77 (M = 53.7, SD = 15.0). These participants were pre-dominantly of White European Caucasian descent (n = 10, 90.1%). Employers included were OEL members for 1–50 years (M = 13.2, SD = 17.0) and had been in the industry for 11–56 years (M = 33.7, SD = 14.7).

The focus groups were moderated by one of the authors (BNK). The groups were audio and video recorded using Microsoft Teams and transcribed verbatim by a professional transcription company. The transcription was validated by one of the authors (ASH) by further review of the audio-visual recording. The study was approved by the Research Ethics Board of the University of Toronto (Protocol #41519).

### 2.3. Data Analysis

Qualitative data were analyzed using an evidence-based, qualitative framework [19] and through a data-driven thematic analysis approach [20]. There was no reliance on a theoretical or pre-existing coding schedule developed by the researchers or the OEL. Initially, the transcripts were examined for inconsistencies and comparable formatting by verifying the transcript with the audio and video file. The transcriptions were reviewed for familiarization with the dataset and to ensure consistency among initial inductive analyses with the “open” coding strategy. Three researchers (JL, SJ, AH) identified codes in the focus group transcription independently using NVivo 12 (version 12.7.0). Notes were shared on codes generated with the first three transcripts, and code labels were generated for independent review of the remaining transcripts. Consensus was reached based on code comparison and validation in research team discussions. After the generation of an agreed-upon set of codes and brief explanations, the researchers discussed any codes that did not fit the themes generated. Key themes were generated by the researchers co-operatively and were subsequently reviewed by the principal investigator to ensure standardization and relevance to the research question. Themes were conceptualized using a semantic approach where explicit meanings in the data were derived directly from the participants’ views. Supporting quotes reviewed in the transcription were collated based on relevance to each theme and associated sub-theme. Impactful quotes were annotated in NVivo 12 to be incorporated in the research findings.

## 3. Results

Analytic review of the focus group data resulted in three overarching and inter-related themes related to this investigation of the effectiveness of the OEL mentorship program. These themes were identified as (a) long-term apprentice recruitment and retention; (b) equity and mental health in the workplace; and (c) summary of industry challenges and mentorship program outcomes. Within these themes, sub-themes were formulated to organize influential factors involved in the participation and engagement rates of apprentices with SMEs. The illustrative quotes presented are denoted and labelled as “FGP”, which is an acronym for “focus group participant”.

### 3.1. Long-Term Apprentice Recruitment and Retention

The re-occurring sentiment across both focus groups was difficulties with the beginning and end of the apprenticeship program for these SME employers, which we have defined as recruitment and retention. Both themes can be used interchangeably given that the intrinsic (e.g., apprentice ratios, unionized labor, wages, disability) and extrinsic (e.g., apprentice age, gender perceptions, pre-apprentice education and diversity) factors that influence the outcome of increasing engagement are interrelated. All the SME employers agreed that increasing non-financial and financial support from the education system and the government is critical to the long-term success of the OEL mentorship program and trades apprenticeships. Some examples mentioned by the SME employers of non-financial support(s) include increasing awareness of trades as a viable career opportunity throughout elementary and high school, collaboration of schools with local employers for co-op programs/student placements and increasing accessibility of vocational resources to consider the apprenticeship program after high school.

FGP1:
*“Now on occasion we get someone who’s 17, 18 years old trying to get into the trade. But they don’t even know, coming out of high school trades are not even really an option, not even promoted as something to do. Or at least that’s what it feels like because we do not get many people that are coming right out of high school.”*


FGP1:
*“And the biggest shortfall is that people are not getting out of high school or even going into high school with a trade even on the horizon. It’s not even in anybody’s mind. It’s I’m going to get out. You know most of the apprentices I’m hiring have gone to two years of college.”*


Some employers had mentioned that they thought the OEL promoted subsidies were helpful in offsetting some costs of the apprenticeship training; however, they also believe there is a lack of financial support from the government to promote, sustain, and encourage pre-apprenticeship training in high schools.

In addition, there was consistent agreement among the SME employers that long-term apprentice retention was a primary concern for non-union employers engaged in the mentorship program. Given the costs associated with mentorship and the training of an apprentice, the employers expressed discontent with the ability to retain apprentices due to external pressures from society, unionized employers, economic instability (e.g., wages and job demand), and apprentice age-related nuances. These concerns are not necessarily critical of the mentorship program per se but rather reflective of years of industry challenges in mentorship inflexibility, a lack of standardization in training, and a lack of government support for the trade industry.

FGP7:
*“In the old day[s] before they had to get registered, there’s a bunch that would have a helper; and then eventually the helper would come on and hire onto me to get his license. And they’d be grandfathered in for a couple of years, or three years because of experience—work experience, and get his ticket in two years and then go out and be his own self-employed in a van.”*


FGP11:
*“It hurts when they leave because it takes—everybody talks about apprenticeship, but for us to have a top gun, that we can use in the industry, it takes me 10 years.”*


FGP8:
*“I think another—another thing that’s missing is back in the day, when we used to hire apprentices, we used to have a Ministry of College rep actually come to our place of employment and meet with the apprentices, sign them up, here’s your program, here’s what you need to do. That’s all gone away. There’s no more social interaction with those ministries, right?”*


FGP8:
*“What I can tell you is that the training and aspects for our apprentices, they never leave. Because we provide them [with] full training through their whole term of apprenticeship. But in the last ten years, retention is becoming far more difficult.”*


#### 3.1.1. Apprentice Ratios

These SME employers have emphasized the opportunity to introduce a probationary screening period to assist with apprentice ratios. During this period, apprentices would be considered as “helpers”, and therefore this would not count towards their apprentice ratio until the employer determined the mentorship candidate was ready for the apprenticeship program. This would also be beneficial to the helper to determine if the employer was the right fit, in terms of workplace culture and mentorship style, and to confirm that the apprenticeship program aligns with their vocational interest. Some examples of this are illustrated below:

FGP3:
*“I get all kinds of requests for people that want to be first term apprentices so my idea would be that along with the ratio is to have someone that is basically designated as a helper and you create 2500[-hour] window or a 3000-h window and that person then becomes just the helper, he’s not registered as an apprentice. He becomes a helper which then helps you [to be] competitive as a contractor. It also helps you basically weed out whether that helper is, you’re willing to include him as part of your ratio, as part of your apprentices, right?”*


FGP1:
*“But that’s why I have the theory that if you had some one that you could take on for about 3000 h right, not count towards your ratio and then if gives you the ability to screen that person for that long whether you want to take him on or not and also gives him the ability to say yeah, this is what I want to do for the rest of my life or no I’m out of here right?”*


Another limitation of the 1:1 ratio for SME employers is that all apprentices despite their years of training in the program are counted equally towards the apprentice ratio. These SME employers argued that there are fundamental differences in the apprentice acumen and workplace productivity for first- or second-year apprentices compared to fourth- and fifth-year apprentices. There was consensus amongst the SME employers that first- and second-year apprentices require full supervision during their apprenticeship activities, whereas fourth- and fifth-year apprentices can perform most work activities independently. The group agreed that flexibility should be introduced for upper-year apprentices with respect to the ratio requirement, but they also acknowledged that may deter employers from recruiting early-year apprentices.

FGP3:
*“And you know if you have six guys on a job, say even if you have four guys on a job site, like he could probably handle once licensed guy and then maybe like another fourth- or fifth-year apprentice and then two newer guys, right? But with the ratios it completely restricts you from doing anything like that.”*


FGP7:
*“I have one gentleman that’s been a fifth term for two and a half years now.”*


Lastly, a recent factor influencing SME participation in apprenticeships from 2020–2022 is the COVID-19 pandemic. Employers are noticing the impact of COVID-19 on the examination backlog of journeyperson candidates to complete their licensing requirements. SME employers reported that these delays are creating additional pressure for employers who are unable to onboard or recruit new apprentices due to constraints of the 1:1 apprenticeship ratio guideline in Ontario. One employer also stated that COVID-19 has also influenced the ability of SME employers to be competitive given supply chain difficulties, resulting in many business owners exiting the electrical industry.

FGP5:
*“We’ve really been noticing this through COVID with the online schooling that some of our apprentices have gone through. They’ve become journey person candidates and then there’s a backlog of testing so they were waiting potentially months to get an exam and if they were unsuccessful then it was an additional month before they could write again.”*


FGP5:
*“They’ve become journey person candidates and then there’s a backlog of testing so they were waiting potentially months to get an exam and if they were unsuccessful then it was an additional month before they could write again. And regardless of the fact that they have completed their apprenticeship, in the eyes of the apprenticeship program they’ve gone and completed all of the schooling, they have all of their logged hours, they’re still considered part of the ratio and our hands were tied to bring on additional apprentices and that just creates a bit of a backlog for new apprentices who are wanting to come on to the program as well.”*


FGP8:
*“…over the last year I can tell you, there’s 5000 less people in the electrical trade, either they’ve closed their business or whatever, other circumstances have happened. But we’re going to have a shortfall.”*


#### 3.1.2. Unionized Labor and Wages

The main contrast between unionized employers and non-unionized employers appears to be greater financial support for union employers; more diverse employee growth opportunities; and ongoing career guidance from trades educational instructors to join union employers. These SME employers highlighted the lower return on investment considering the lack of financial support for non-unionized employers in training an apprentice compared with the time and resources required to train them. Some excerpts from the focus group describe this further and mention the limitations of competing against unionized employers:

FGP7:
*“Well, why would a private contractor invest $30,000 in training an apprentice for the union to poach them?”*


FGP6:
*“We’re just a training center for the union half the time. I cut back and get five on payroll now, but what happened was the union was dictating to the non-union contractor what we could do and what we couldn’t do.”*


FGP3:
*“You’ve got to teach them everything and then you risk losing them whether you lose them to a union, another company or just going into another field.”*


Another consideration is that currently the cost of training apprentices and the risk of poaching significantly outweigh the benefits. Many of the SME employers mentioned that it would take a couple years for the apprentice to become sufficiently trained to be productive for the employer. Larger, unionized employers have utilized poaching as a strategy to acquire well-trained apprentices by paying them higher wages rather than absorbing the initial training costs of a 1st or 2nd-year apprentice.

FGP11:
*“How many in industry don’t bother training? They go to the market; they go to our people that we have trained to hire them away. That’s happening, uh, all the time to us…. They said, uh, the people from hydro come along after they’ve we’ve done the training, offer them more money and so, they say we’re not going to train anymore. Do I really want to train somebody? Do you think half the contractors don’t train any apprentices, they just say, “I’ll wait until somebody’s trained, and I’ll offer more money?” “Of 9000 electrical contractors, we know how many are not training any apprentices, maybe half.”*


FGP6:
*“The other unfortunate part is the, what I find as a non-union contractor is I find it very hard now as they jack minimum wage up, we work under percentages of what your rate of your, what your shop is.”*


A secondary pressure on wages is economic volatility through inflation and cyclic supply and demand constraints that can result in work shortages or layoffs. This is also influenced by increases in the minimum wage in non-urban areas of Ontario despite stable or slowly rising service costs.

FGP6:
*“So, there’s a training aspect here that every time they jacked the minimum wage, everybody’s wages have got to go up and it’s becoming harder and harder.”*


FGP6:
*“…we should be getting subsidized as the minimum wage was up because it’s not, the farther they drive the minimum wage up it’s just going to be, it’s more ridiculous for the people that you already have on senior people on staff because an area will only take what an area can be charged out at. You can’t start charging where I am in a rural Kingston area.”*


FGP1:
*“I think it’s next to impossible to hire anybody right now, it’s a very, very difficult market and wages are increasing exponentially, faster than we can definitely put up our rate.”*


#### 3.1.3. Apprentice Age

Throughout the focus group, there was a stark difference in retention and engagement rates amongst the SME employers with apprentices aged 18–24 compared to those that were 25–29 years old. Most of the SME employers reported that there was difficulty retaining apprentices aged 18–24 due to generational gaps in work ethic (e.g., punctuality, interpersonal communication) and information access (e.g., social media, cellphone usage). They had some concerns with the responsibility of younger apprentices stating that they are not prepared for success in the trades industry with respect to tool handling, material knowledge, and employee work etiquette. This lack of preparedness was largely attributed to the educational system not promoting students to consider the trades and lack of exposure to preparatory workshops/courses during high school. The consensus among the SME employers was that older apprentices (>25 years old) had more life responsibility and therefore were more desirable given their productivity, reliability, and higher retention rates during the trades apprenticeship program.

FGP3:
*“There’s not too many guys that will hire first term apprentices especially 18, 19-year-old kids coming out of high school.”*


FGP7:
*“But when you’re talking to a person who is 21 or 22 years old, they don’t have the ability to perceive the value of working with the company for 20 plus years, having the same people to work with, a stable secure environment. Um, because, once again, it goes right back to when I talked about starting to expose people in grades seven and eight to these ideas.”*


FGP1:
*“…it has to start Grade 9, Grade 10. When someone’s now working toward what potentially they’re going to do for their life. If it starts after that it’s definitely not too late. You’re not too old to get into a trade at 25. I mean most of the apprentices I’m hiring are 25 years old. But I think if we want to fix this program, if we want to fix how the trades work, it’s got to start at 14, 15.”*


#### 3.1.4. Pre-Apprentice Education

SME employers stated that over the last decade there have been generational shifts in educational pursuits that have evolved from advancements in technology. This pivot in the educational system has impacted the trade industry through recruitment, retention, and student perception of the viability of a vocational pursuit in the trades industry. This student perception is further reinforced by parental and societal attitudes of “blue collar” vs. “white collar” jobs as seen below:

FGP10:
*“We haven’t had a single person, um, go through that program that’s decided to become an electrician.”*


FGP7:
*“…I have been to all the high schools, I have been to the OEL nights, I’ve been to the Georgian College meet the kid’s night, we’ve set up trade fare booths all in the light of promoting. But we never get the key to talk to the teachers, or to talk to the principals, to talk to the directors in the boards of education to make decisions of the curriculum for the schools. And I think that’s where some core—um—I don’t know what the right word is—training—but that’s not the right word, put aside the second-class nature that this is all interpreted as starting at grade nine or grade eight or grade seven.”*


FGP8:
*“But, you know, the teachers in the school system still don’t really promote it. And the parents think of it as second class. And you know, the kid’s uncle that’s a professional something or other, you know, you don’t want to go be an electrician.”*


SME employers also stated that most of the apprentices they have trained are not prepared for the apprenticeship program as they have minimal workplace etiquette or have no general knowledge of the tools or materials used in the industry.

FGP10:
*“Um, through high school, we probably get, oh, I would say probably two or three students a year that are interested in the electrical trade. The biggest problem with that, we have about—with all those, we’ve probably been doing it for probably ten years with the high school, I think we’ve only hired one person in that whole ten years that was going to be somebody that would make it as an electrician.”*


FGP9:
*“[I] teach at a college level too, and I can tell you based on what you said earlier, I literally have you know 23-year-old students coming in who do not know what end of a hammer to hold. They’re so excited when you actually teach them how to operate a drill, uh, when you actually you know start explaining wire sizes to them.”*


### 3.2. Equity and Mental Health in the Workplace

Some of the SME employers had mentioned that the worksite conditions have improved over the past couple of decades, but they believe there remains further considerations to achieve sustainable participation of diverse individuals in the electrical trades.

FGP5:
*“One of my areas of focus right now is around mental health in the work place and it pains me to hear some of the stories and see some of the situations on construction sites on how individuals are spoken to. Having electricians, having apprentices move over from other companies and just the amount of pressure that has been put on some of these individuals from a mental health perspective.”*


In the trades industry, the SME employers stated there has been increased societal stigma that has prevented long-term growth in recruitment and retention. Teachers and parents motivate young people into “professional careers” based on financial viability and into careers with exponential long-term growth opportunities. Careers within the trades have been largely considered an alternative when you are considered not successful within the traditional, hierarchical education system. The general exception to this, as illustrated by the focus group participants, was those who had a family history of trade workers.

FGP1:
*“Yeah, I just wanted to continue one with the OAP and high school. I think I definitely agree with you, there’s always been a stigma. I talked to my dad who, previous generation of our company, [is an] electrician, our previous master electrician. Trades is for the dummies, that’s the comment that he says when he was going to high school and I don’t know that it’s changed drastically.”*


FGP8:
*“So, I think there’s a stigma with–you know between blue collar and white-collar workers. And you know when you’re considered a blue-collar worker everybody kind of like downgrading you all the time. And for us as a company we’ve been here in business now for over 30 years.”*


FGP1:
*“And you know, basically there is that aspect, the hard part of the job and you can’t have somebody that’s 80 or 90 pounds carrying two rolls of wire on their shoulders especially when they’re not built for it. So, there is that stigma too I guess that women can’t do this.”*


FGP4:
*“And they actually feel like they’re less than what they really are as people right? And what I tried to explain to them is that they just have a different skill set as compared to those other children, with those other kids and that they do and will provide something positive to society, becoming a skill[ed] tradesperson.”*


#### 3.2.1. Gendered Perspectives and Diversity

These SME employers have reported mixed findings with recruiting female apprentice candidates. Some SME employers have reported success recruiting females through co-operative programs at their local secondary school while others reported that they have seen very few females at their workplace or worksite, and if they do hire women, there are significant dropout rates.

FGP6:
*“…girls in high school are coming forward because I’m finding that now with myself with the co-op students.”*


FGP2:
*“It’s nothing against saying that women cannot go into the field, it’s just not common for me in my case because I have only seen one person, one female in a job say in 15 years.”*


However, there was consensus amongst the group that there is openness to recruit women to the electrical trades; however, there are concerns about capacity to endure the physical nature of the job and potential frequent exposures to physical hazards. Some employers within the group have re-structured their business to re-evaluate their workplace culture and create positions within the industry to allow for gender-neutral adaptability.

FGP5:
*“We do not have very many female electricians apply when we have job openings which is unfortunate. I would love to see more women in the industry. It’s a massive industry and I think that there’s room for many more unique individuals to enter the industry. So, there may be some areas that women would love to be in the electrical industry and have that conversation and say you know, I’m not interested in these types of jobs.”*


FGP5:
*“So, we’ve done a lot of restructuring in our business to really promote and to be really intentional about the clients that we will work with to protect the mental health of our employees. And I think that maybe that is part of the gap [because] there are not as many women involved…”*


FGP10:
*“It wasn’t because they were female that we hired them, it was solely because of their abilities.”*


Despite this, these SME employers identified some industry-wide barriers that prevent an inclusive environment such as a lack of gender-inclusive safety protocols and women-designed personal protective equipment (e.g., boots and overalls), employee promotion of a masculine, self-reliant environment (e.g., disapproval for asking for help), and workplace accessibility issues (e.g., having sanitary, gender-neutral washrooms). They stressed these considerations need to be applied industry wide and not just at the SME employer level to observe the intended effect of these changes. They viewed these barriers as creating a hegemonic masculine environment and could be viewed as a deterrent for women to consider a career in the electrical industry as there can be a lack of belongingness.

FGP4:
*“I think the toughest part about being a woman in any construction industry is the mental aspect and how they’re viewed and how they’re treated and treated by other trades people, right?”*


FGP1:
*“…we just hired two women recently. Both of them in the interview when asked them what are your expectations or what are you looking for in this job, their first, both of them their first answer was just an accepting environment.”*


There was widespread openness amongst the SME employers to willingly hire those from underrepresented groups (e.g., Indigenous, non-white ethnic backgrounds, newcomers to Canada) in the electrical trades. Retention of these diverse groups was mentioned to be difficult given that many of the non-financial supports are not readily accessible or accommodating (e.g., language barriers) to these populations. For example, an SME employer mentioned that many newcomers to Ontario have been trained differently with different materials and tools in their country of origin.

FGP8:
*“I think another component of that is you know we try to promote anybody that comes into the trades. However, a lot of them are constricted by language and unfortunately, a lot of the codebooks and a lot of the training manuals, uh, that you do go through your apprenticeship with, they’re either written in either English or French I believe. But mostly English. You know you have to start converting some of your manuals to accommodate you know the new people coming into the trades.”*


FGP9:
*“If you’re not exposed to it, it’s not something you’re ever going to think about. So, if you don’t see anyone that looks like you in the trades it’s not something you’re going to want to do. If everything that’s written or everything that’s public about that trade is in a language that you don’t understand you’re not going to be interested in it.”*


#### 3.2.2. Workplace Culture

There was consensus amongst this group of SME employers that there needs to be a shift in employer culture to make the construction electrical industry more inclusive. There is agreement that the current employer culture is based on long-standing traditional principles passed down from generations-to-generations of male electricians. This can result in difficulty in recruiting new groups to the electrical trades as they may not thrive in an outdated, male-dominated workplace environment. The upside to this is that new groups and diverse role models in the electrical trade can help innovate the culture to support new entrants into the industry.

FGP1:
*“Going to a job site makes it a little more difficult because you’re dealing with other sub trades and contractors and other people that might have a different mandate, but I don’t, I mean I don’t think you’re going to change the 50-, 60-year-old tradesman, or trades person, tradesman we should say because most of them are men. You’re not going to change that mindset. You have to start with the new people coming in and change the culture there.”*


Many of the SME employers in this group have mentioned they have implemented increased accountability with their apprentices and journeypersons for how they present themselves on the worksite. If more employers focus on improving their internal culture, ultimately it will promote a more inviting apprenticeship program and allow apprentices to find value within the program.

FGP3:
*“Well going back to the mental health issue, I think it is our job to create good cultures in our companies and change that it is on the construction field. Again, in my case, how I said I didn’t let anybody yell at me, it’s not like I walk away from a job site and say you know, I’m not taking this. I have situations when someone tried to be disrespectful to me, but I let them know that I’m not OK with that and I’m not going to be willing to take it from that point on.”*


FGP5:
*“And it’s not OK that they talk to you like that. You’re a human being as well. If they can’t talk to our office staff in a certain manner, there’s no way they can talk to you as a foreman in that manner as well. And I think that that is a huge gap in the industry as far as employability goes because the reality is that people who are coming in to the industry, they don’t have the same tolerance and leniency to be treated the way that historically some of these constructions trades have just gotten away with, to be quite frank.”*


### 3.3. Summary of Industry Challenges and Mentorship Program Outcomes

Despite the factors (e.g., recruitment and retention) influencing participation and engagement of SME electrical employers, this group of employers is optimistic for the future and has provided recommendations on how to improve some of the challenges in the industry. Many of the SME employers are long-standing mentors in the OEL and continue to promote the apprenticeship program, as well as utilize the employer support resources available through the OEL.

Small-to-medium electrical employers have significant supplementary costs when providing mentorship to an apprentice. These costs include materials, tools, supervisory costs, and basic standardized training costs (e.g., safety training, circuit training, alarms). Many of the employers within this group agreed that they need further support to streamline basic training while providing mentorship.

FGP11:
*“We got to teach safety, for example, to all our people before they start. We don’t get any assistance in that, that’s a big cost, uh, to do the safety and then somebody leaves.”*


FGP4:
*“Some technical background and it would be like a Saturday morning style class similar to what the union does for some of these apprentices, just to give additional training whether it be fire alarms, circuits. You know, simple things. It’ll just grow, it’ll help them grow as electricians down the road.”*


The OEL and its members have been working on lobbying the government to create training centers to allow for collaborative learning of safety and basic electrical training based on feedback from OEL member employers. Currently, many of the SME employers as members of the OEL operate on a volunteer basis and may not be able to dedicate the time and resources to contribute to establishing the training centers. By allowing these SME employers to focus on creating a quality mentorship and inclusive environment, they can avoid the external pressures of poaching from the larger employers in the electrical industry.

FGP4:
*“I would think that it would be nice if they had additional training even if it’s once a month specific training for some of these apprentices.”*


When discussing improvements to the apprenticeship program, many of the SME employers were reflective of previous years when there was more involvement with the provincial government. They appreciated in-person contact and bi-weekly check-ins from the government ministries to ensure that they felt supported throughout the mentorship. COVID-19 has certainly impacted recruitment and retention for many of these SME employers, as many communications are now electronic with many candidates arriving on the worksite without having met the employer in person. There is a shared sentiment among the SME employers that the journeyperson camaraderie of these smaller businesses lends to a preference of in-person social interactions over electronic communications. Their general perception is that they would be able to select higher quality candidates if they were able to meet them in-person and engage with them.

FGP8:
*“We used to have a Ministry of College rep actually come to our place of employment and meet with the apprentices, sign them up, here’s your program, here’s what you need to do. That’s all gone away. There’s no more social interaction with those ministries, right.” That makes a difference that you matter you’re valued.”*


The SME employers provided positive feedback on the OEL’s support of employers participating in the mentorship program. Many of them praised the onboarding process, and hiring tools have provided positive experiences for these employers when they are recruiting apprentices to the electrical trades. The accessibility and resourcefulness of the OEL representatives have allowed these employers to not only be supported at the beginning but also through the entire 5-year apprenticeship.

FGP5:
*“I have had really great experience with their job boards and hiring through the OEL talent sorter. ‘I believer’ is who they are partner[ed] with so I found that to be a really great resource because hiring in the electrical field can be difficult as well. We have a fantastic representative that’s been very supportive with any questions that I’ve had onboarding new apprentices.”*


FGP9:
*“The boot camps are designed to go across to all the high schools, or I believe it’s high schools; not just high schools. But it gives them a taste of what the electrical trade is like. And if that interest is there, then that’s when those apprentices get funneled over to the OEL. And that’s when we take over as mentorship and trying to help them further get them involved with a potential employer that want to hire them.”*


There was one critical feedback about the apprenticeship program regarding sponsorships:

FGP5:
*“One piece of the process that I’ve always found extremely awkward is transferring of sponsorships. So essentially if we hire on a third or a fourth or a second whatever apprentice, we require a letter from their previous employer confirming their hours, which is great in a perfect world, but if they’re unable for whatever reason to provide that or they’re estranged from their previous employer for whatever reason, we haven’t run into it but I’ve always kind of thought in the back of my head, it puts people in a really awkward situation to ask for this.”*


## 4. Discussion

Continued leverage of engagement and support of SME employers in the construction trades is an important avenue for growth to address the ongoing shortage of skilled trades workers in Ontario, Canada. In this study, a subset of SME employers engaged in a non-for-profit industry-led mentorship program provided by the OEL. These employers appreciated the value of the mentorship program through praise of the continued educational support, employer management expertise, hiring resources, and onboarding tools. They did identify some barriers to growth in the electrical industry but had a positive outlook for continuing to foster equitable apprenticeship training opportunities for those interested in the electrical trades industry. There were some key areas of concern discussed by the employers regarding apprentice retention, recruitment, and mental health outcomes related to workplace culture and trade stigma.

Apprentice retention continues to be difficult given the impacts of the COVID-19 pandemic, economic changes, and changes in societal interests [21,22]. In this study, these employers had mentioned anecdotally that many businesses have had to close or transition out of the industry due to the pandemic. The full impact of the pandemic on the construction trades industry for electrical contractors is yet to be fully known, but some research groups have projected an increase in the severity of psychosocial stressors and burnout due to social and professional isolation [23] as well as a disruption of work–life balance [24]. The pandemic has also compounded the influence of changes in economic activity and financial resilience that have presented an additional challenge to SME employers engaging in creating mentorship opportunities. Many of these employers continue to advocate for these opportunities but do not have the financial resiliency of larger unionized employers to lose apprentices to inter-industry attrition or intra-industry poaching [3,21,22]. In addition, the pressures of journeyperson wage increase in non-urban areas have caused a reduction in business productivity (e.g., revenues, asset-to-debt ratios) on their ability to focus on training and retaining high-quality apprentices. This is problematic in that many of the employers that participated in the focus groups agreed that it takes upwards of 10 years to recruit and train a highly productive, well-trained apprentice to contribute to net business productivity (as defined as profit after training expenses). This perception remains reinforced by the fact that pre-apprenticeship training is critical to the value proposition of the employer–apprentice relationship for small-to-medium-sized employers. Employers in apprenticeship programs of other countries have supported this proposition and observed it to be associated with increased registration and completion rates in the construction trades [25]. Therefore, to increase apprentice retention, employers must be supported with comprehensive pre-apprenticeship programs at the government level and through its community constituents (e.g., schools, community centers, non-for-profit industry advocates) [21,26,27].

Given the correlation between the factors that influence both recruitment and retention, employers also have a social responsibility to foster relationships through ongoing vocational outreach with school-aged students to encourage further engagement with the construction trades industry [3,27]. This social responsibility can be illustrated through efforts to reduce generational disparities between SME employers and apprentices by improving interpersonal communication, transference of industry knowledge, and perspective sharing regarding their vocational experience as prior apprentices and journeypersons [26,27]. A more supportive approach will allow younger apprentices (<25 years of age) to be able to evaluate the many intrinsic (e.g., personality style, temperament, attitudes/perceptions of career choice) and extrinsic (e.g., finances, future career growth and longevity, health) factors that influence their vocational decision-making as they transition from societal to individual responsibility after high school [26,27]. Recent studies have also suggested this with greater rates of dropout and dropout consideration in the construction industry for males and females aged 18–24 compared with their 25–64-year-old counterparts [12,13,28]. On the other hand, SME employers in this study and other studies on the construction trades [25] believe that individual motivation, work ethic, and general construction knowledge are desirable apprentice attributes for employers looking to recruit an apprentice. These attributes have also been subjectively reported with greater retention and positive vocational outcomes. For a successful and equitable relationship between apprentice and employer, employers must look to reduce information and hands-on learning barriers through stronger, interactive partnerships with younger apprentices and the institutions that support them.

The recruitment of underrepresented persons (e.g., women, newcomers to Canada, persons with disabilities) has also been presented as a key consideration to encourage the further acquisition of vocational talent outside of the prototypical hegemonic masculine applicant [11,29,30]. Diversity and inclusion policies have been proposed for this industry to engage young under-represented persons previously, but it is currently unknown which mechanisms would promote the sustainable engagement of underrepresented persons [31,32]. For example, it was observed in a survey study that companies with 21–50% of female employees had dedicated recruiting and retention practices for women [26]. The authors mention that the direction of this relationship was unclear, and therefore, it is unknown whether this is an application of critical mass theory or organizational-driven policy and procedures [12,13,31,33]. Employers may also have to be cognizant of the effects of workplace interest and anxiety when recruiting underrepresented individuals [25,34]. Workplace environments in the electrical and plumbing industries can be reflective of a hegemonic masculine stereotype (e.g., characteristics of physically strong, competitive risk-taking behaviors, white Caucasian, and heterosexual), which may reverse the effects of initial high workplace interest with greater workplace anxiety in underrepresented persons [35]. In this study, the SME employers expressed an openness to recruit underrepresented persons but also cautioned about workplace (and worksite) culture, health and safety hazards, and the physicality or “self-reliant” of being an electrician or plumber, which is consistent with the construction trades literature [9,36]. One study indicated that higher levels of dropout were associated with apprentices reporting feelings of verbal abuse, harassment, or workplace exploitation [25]. Further research is required to determine the influence of mental health training or workplace culture interventions on recruitment and retention outcomes [37,38]. Nonetheless, further integration of under-represented persons may allow for the re-evaluation of current organizational policies to create adaptations in health and safety policies, the revision of current job descriptions, or the creation of new jobs to accommodate diverse apprentice strengths that are unrelated to physicality and self-reliance [8,9].

There are few studies investigating the roles of workplace culture and construction trade stigma as it relates to employer or apprentice mental health symptoms and psychosocial outcomes [38]. There was some commentary among the SME employers in this study that workplace culture has improved over the years, but there were identified barriers preventing integration for underrepresented persons previously discussed in the literature related to workplace facility accessibility (e.g., gender-neutral washrooms, open and inclusive environment) [39], access to personal protective equipment designed for different body types and genders [36], and specific task assignments to females as compared to males [28,40]. In the literature, there is an increased incidence of mental health symptoms (e.g., sleep problems, anxiety, and depression) [38], and negative coping mechanisms (e.g., workplace verbal abuse, aggression, social isolation, substance abuse and dependency) in construction trade workers may be indirectly related to their workplace culture and the physical demands of their day-to-day work resulting in musculoskeletal and psychological strain [41,42,43]. Studies have also shown that men experience higher employment-related stress but are less likely to engage external mental health supports due to mental health stigma [38,43] and therefore are vulnerable to negative mental health outcomes associated with workplace accident(s), injuries, and fatalities [35,37]. Given these negative outcomes, a few studies have explored passive and active interventions to develop awareness, reduce stigma, and build psychological resiliency [2,34,44,45]. A suicide prevention training program implementation study reported longitudinal improvements in help-seeking behavior and reduced mental health stigma in a sample of Australian construction contractors [45]. No significant differences in the administration of the training (e.g., formal and informal) and high attrition rates observed may limit the findings of this intervention study. Further research is required to understand the effectiveness of the training, disentangle the effects of the components of the training, and determine whether this could be applied to other mental health interventions across different construction-related industries.

Several limitations have been identified in this study. Qualitative research, in general, can contribute to the development of a conceptual framework to support further investigation in an area of interest. Therefore, the absence of quantitative data or other study elements may reduce the generalizability of the data presented in this study and may not be an accurate representation of the effectiveness of the mentorship program in totality. The sample size is small for a qualitative study, and therefore, thematic conclusions derived from this sample should be considered explorative in nature. The demographic composition of the sample may not be representative of the electrical trades industry as there was only one underrepresented employer and sampling was limited to employers that voluntarily participated in the focus group. Lastly, the virtual aspect of the focus groups may have affected the group dynamics of this sample since many employers within the groups mentioned the preference for in-person meetings.

## 5. Conclusions

For a large portion of the global workforce, mental health and work are intimately connected. This is no different in the skilled trades, and the findings of the study indicate that SME employers had a positive, supportive experience participating in the OEL mentorship program. Many of the employers indicated that they would promote the mentorship program to others and continue to engage in OEL events/training. On the other hand, there remain SME employer challenges in recruitment and retention that were identified and not able to be addressed within the mentorship program related to equitable engagement of underrepresented persons, pre-apprentice development, and mental health in the workplace resources. As such, we have some recommendations to refine the OEL mentorship program and other SME employer-driven support programs in Ontario. Industry partners and other key informants should continue to focus efforts in strengthening these key areas to promote sustainable growth in the construction trades industry: (a) enhancing pre-apprenticeship development programs and improving relationships with the community by matching local employers and high schools; (b) continued funding and supportive programs to increase pre-mentorship opportunities for young people to engage in and be exposed to the day-to-day activities of an electrical journeyperson; (c) reducing trade stigma and attitudes towards the inclusion of underrepresented parties in the younger generations will drive future growth in registration in the construction electrical trades; (d) to address the current hegemonic masculine workplace environment, there needs to be the development of policies that reflect the equity of underrepresented persons and training to promote social assertiveness in the workplace. Industry partners should work with SME employers to develop workplace initiatives and engage external partners to provide ongoing apprenticeship mentorship support to address the recruitment and retention barriers identified in this study.

## Data Availability

The data presented in this study are available on request from the corresponding author. The data are not publicly available due to privacy.

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
