# Peer review of "Engaging Employers in Apprentice Training: Focus Group Insights from Small-to-Medium-Sized Employers in Ontario, Canada"

_ijerph, 2023, doi:10.3390/ijerph20032527_

Round 1

Reviewer 1 Report

A very well written paper. The methodology could have been strengthened with more quantitative data. It was surprising that the experience of apprentices with their supervisors was not mentioned as a theme. In other research around apprentices the relationship and supervision factor has been seen as a significant factor in retention.

Author Response

Reviewer Rebuttal

Reviewer 1: A very well-written paper. The methodology could have been strengthened with more quantitative data. It was surprising that the experience of apprentices with their supervisors was not mentioned as a theme. In other research around apprentices, the relationship and supervision factor has been seen as a significant factor in retention.

Response: Thank you for your feedback, we appreciate your comments. We agree the methodology could be strengthened with quantitative data; however, this is a qualitative study, and we did not want to deviate from the appropriate methods for this design. We agree that the relationship with supervisors can be identified as a significant factor in retention. However, during the focus groups with participants, not enough information was collected on this topic to explore this factor.

Reviewer 2 Report

The purpose of the study is to evaluate the “efficacy” of the Ontario Electrical Leagues' mentorship program through the perspectives of small-to-medium (SM) sized employers. Since the SM enterprises are weak in supporting their employees, it is important to clarify how apprentice training programs can be improved.

However, there are limitations in this study in several aspects. First, the reviewer suggests that qualitative research can only contribute to the development of a conceptual framework or research questions on one focus area but it is difficult to evaluate the efficacy of the program.

Second, the authors need to define the core concepts of the study for clarity, because they explore a multidisciplinary field between labor policy and public health policy. Examples of core concepts include SM enterprises and mental health at work.

Third, the authors should re-check the consistency of the meanings of the term “mental health.” Participants in the study used “mental health” in a broad sense, whereas the authors might use “mental health” in a more sophisticated or professional way. The reviewer recognizes the gap in these two meanings.

Lastly, related to the third point, it is difficult to follow why the authors concluded by highlighting the importance of mental health consultation and resources in the final sentence. This is partly because the authors mentioned the limited scientific evidence on mental health interventions in the discussion, and partly because it is unclear what “the mental health consultation and resources” mean.

Author Response

Reviewer Rebuttal

Reviewer 2: The purpose of the study is to evaluate the “efficacy” of the Ontario Electrical Leagues' mentorship program through the perspectives of small-to-medium (SM) sized employers. Since the SM enterprises are weak in supporting their employees, it is important to clarify how apprentice training programs can be improved.

However, there are limitations in this study in several aspects. First, the reviewer suggests that qualitative research can only contribute to the development of a conceptual framework or research questions on one focus area, but it is difficult to evaluate the efficacy of the program.

Response: Thank you for your comment. We believe the qualitative approach of this study evaluates one aspect of the program through the in-depth perspectives of SME employers. We acknowledge this as a limitation and have added this to the discussion section.

Second, the authors need to define the core concepts of the study for clarity, because they explore a multidisciplinary field between labor policy and public health policy. Examples of core concepts include SM enterprises and mental health at work.

Response: Thank you for your comment. We have defined "SM enterprises" and "mental health at work" and other core concepts discussed in this manuscript, accordingly.

Third, the authors should re-check the consistency of the meanings of the term “mental health.” Participants in the study used “mental health” in a broad sense, whereas the authors might use “mental health” in a more sophisticated or professional way. The reviewer recognizes the gap in these two meanings.

Response: Thank you for your comment. We have reviewed the consistency and usage of the concept "mental health" ensuring there was a delineation between participant usage and our usage throughout the manuscript.

Lastly, related to the third point, it is difficult to follow why the authors concluded by highlighting the importance of mental health consultation and resources in the final sentence. This is partly because the authors mentioned the limited scientific evidence on mental health interventions in the discussion, and partly because it is unclear what “the mental health consultation and resources” mean.

Response: Thank you for your comment. We have reviewed our concluding statements and revised them based on our study findings.

Reviewer 3 Report

This study provides qualitative findings on barriers to employer recruitment and retention of electrical apprentices in  the Ontario construction trades.  The work is important and findings are informative.  Below are my recommendations for strengthening the manuscript.

Introduction

Information on who participates in apprenticeship these programs and where they fit into the education/training system in Ontario would be helpful. This section focuses more on just the limits of these programs but it would be helpful to have a bit more of a general sense of how these programs work/ or should work.

Methods

On page 3, lines 100-102 ish…the wording uses the past tense but then on the next few lines, the tense changes.

On page 3, lines 107-113 includes information about the OEL program and should be integrated into the introduction.

The authors write that the focus of this paper is on electrical and plumbing employers.  However, if 10 of the 11 focus group participants represented electrical employers, this paper should not be presented as including findings about the plumbing trades.  This should be reframed as a paper about the electrical trades.

The authors should provide their rationale for defining small-medium enterprises as they did. There are no citations or justification for this definition.

The authors refer to focus group participants as “employers” but it is unclear what position these people hold in the companies they represent.  They should clarify whether these are owners, managers, or what exactly their role is in their companies.

Under data analysis, the authors describe the analytic method they used by referring the reader to a “qualitative framework as described in previous methodological research” with citation provided.  There needs to be much more explanation provided on this approach and the methods used in this work.

Results

Too often, the paper’s headings and subheadings are not reflective of the content under them and often they are vague.  Additionally, there may not be a need to divide up the findings so finely as has been done. Often, the same issue is discussed as a barrier in multiple subheadings. For example, recruitment, retention issues are repeated over and over under different headings. Interest in the trades is also something that is repeated in different places.  These are just two examples. It might be useful to identify just a few major themes and put all similar findings under these. A real reorganization of themes appears to be needed.

Authors should be more cautious throughout about making statements that are definitive. Sometimes there are sweeping statements made about a finding with only one supportive quote. Pulling single quotes under a larger topic might help. Additionally, the authors should be more cautious about making recommendations on the strength of the evidence put forth in the results section.

In the results section, the authors often begin each subsection with a introduction of sorts. These introductory paragraphs read like literature reviews or more often, opinions because there are no citations.  Sometimes, they begin with actual findings.  It is unclear sometimes, whether the authors are giving their opinions or if they are citing the literature, or if they are referencing findings from the focus groups.  All of this makes the findings difficult to digest. Some of the text feels like it would be better either in the introduction or discussion section.

In the section on apprentice ratios, the authors should explain this concept further for those who are not familiar with these policies.  As in other places, the discussion of trade stigma reads like a discussion rather than a finding. Authors should reserve their observations for the discussion section and not put their views in the results.

An editor should review the work to improve structure, English use, grammar, etc.

Discussion

This section is stronger than the rest of the paper.  The writing and organization is well done and statements are consistently supported by citations. 

Conclusions

The first statement of this section is not supported by the results presented in this paper.  I was surprised when I read the first sentence as the paper included very little in the way of the views employers had of the OEL program.  The entire paper was about barriers to recruitment and retention and other barriers to hiring and keeping a skilled workforce.  There were no findings presented with respect to what works and does not work with the OEL program.  This section should be rewritten to focus on the barriers as reported in the bulk of the paper.

Author Response

Reviewer Rebuttal

Reviewer 3:

Introduction

Information on who participates in apprenticeship programs and where they fit into the education/training system in Ontario would be helpful. This section focuses more on just the limits of these programs but it would be helpful to have a bit more of a general sense of how these programs work/ or should work.

Response: Thank you for your comment. We have now added further information to explain further the apprenticeship model and how the OEL program is designed to support the apprenticeship model.

Methods

On page 3, lines 100-102 ish…the wording uses the past tense but then on the next few lines, the tense changes.

Response: Thank you for addressing this. We have made this change to ensure consistency.

On page 3, lines 107-113 includes information about the OEL program and should be integrated into the introduction.

 Response: Thank you for your comment. We have integrated this information in the Introduction alongside the information regarding the apprenticeship model in Ontario.

The authors write that the focus of this paper is on electrical and plumbing employers.  However, if 10 of the 11 focus group participants represented electrical employers, this paper should not be presented as including findings about the plumbing trades.  This should be reframed as a paper about the electrical trades.

Response: We agree with this comment. We have now specified that this manuscript is specifically about electrical trades employers.     

The authors should provide their rationale for defining small-medium enterprises as they did. There are no citations or justification for this definition.

Response: Thank you for bringing this to our attention. We have provided a citation for this definition. 

The authors refer to focus group participants as “employers” but it is unclear what position these people hold in the companies they represent.  They should clarify whether these are owners, managers, or what exactly their role is in their companies.

Response: Thank you for your comment. We have identified the role of employers included in this study between lines 70-76 on page 2.

Under data analysis, the authors describe the analytic method they used by referring the reader to a “qualitative framework as described in previous methodological research” with citation provided.  There needs to be much more explanation provided on this approach and the methods used in this work.

Response: Thank you for your comment. We have now added further details on the methodological process applied in this study.

Results

 Too often, the paper’s headings and subheadings are not reflective of the content under them and often they are vague.  Additionally, there may not be a need to divide up the findings so finely as has been done. Often, the same issue is discussed as a barrier in multiple subheadings. For example, recruitment, retention issues are repeated over and over under different headings. Interest in the trades is also something that is repeated in different places.  These are just two examples. It might be useful to identify just a few major themes and put all similar findings under these. A real reorganization of themes appears to be needed.

Response: Thank you for your comment. We acknowledge the overlap between recruitment and retention factors influencing the electrical trades. We have reviewed the themes once again and have tried to be more cohesive and concise in our approach. Accordingly, we have condensed our themes to (a) long-term apprentice recruitment and retention challenges; (b) equity and mental health in the workplace; and (c) industry challenges and mentorship program outcomes. We have also reduced the amount of sub-themes to reflect greater strength in the themes and sub-themes organized.

Authors should be more cautious throughout about making statements that are definitive. Sometimes there are sweeping statements made about a finding with only one supportive quote. Pulling single quotes under a larger topic might help. Additionally, the authors should be more cautious about making recommendations on the strength of the evidence put forth in the results section.

Response: Thank you for your comment. We have reviewed the manuscript thoroughly to ensure we have avoided any definitive statements and strengthened the empirical data to support an inference.

In the results section, the authors often begin each subsection with a introduction of sorts. These introductory paragraphs read like literature reviews or more often, opinions because there are no citations.  Sometimes, they begin with actual findings.  It is unclear sometimes, whether the authors are giving their opinions or if they are citing the literature, or if they are referencing findings from the focus groups.  All of this makes the findings difficult to digest. Some of the text feels like it would be better either in the introduction or discussion section.

Response: Thank you for your comment. We have reviewed this recommendation and have applied the necessary changes to make the findings to increase readability.

In the section on apprentice ratios, the authors should explain this concept further for those who are not familiar with these policies.  As in other places, the discussion of trade stigma reads like a discussion rather than a finding. Authors should reserve their observations for the discussion section and not put their views in the results.

Response: Thank you for your comment. We have reviewed this and amended it accordingly.

An editor should review the work to improve structure, English use, grammar, etc.

Response: Thank you for your comment. We have reviewed the manuscript thoroughly to ensure that there is a high standard of writing throughout. 

Discussion

 This section is stronger than the rest of the paper.  The writing and organization are well done and statements are consistently supported by citations.

Response: Thank you for this feedback.

Conclusions

The first statement of this section is not supported by the results presented in this paper.  I was surprised when I read the first sentence as the paper included very little in the way of the views employers had of the OEL program.  The entire paper was about barriers to recruitment and retention and other barriers to hiring and keeping a skilled workforce.  There were no findings presented with respect to what works and does not work with the OEL program.  This section should be rewritten to focus on the barriers as reported in the bulk of the paper.

Response: Thank you for your comment. We have decided to revise this section to accurately reflect the data derived from the focus groups.

Round 2

Reviewer 2 Report

The reviewer suggests to the authors that they include the WHO report in the reference.

https://www.who.int/publications/i/item/9789240053052

Author Response

Thank you for the suggestion. The reference has been added to the manuscript